# Adiponectin and TNF-Alpha Differentially Mediate the Association Between Cystatin C and Oxidized LDL in Type 2 Diabetes Mellitus Patients

**DOI:** 10.3390/ijms26073001

**Published:** 2025-03-25

**Authors:** Ahmed Bakillah, Ayman Farouk Soliman, Maram Al Subaiee, Khamis Khamees Obeid, Arwa Al Hussaini, Shahinaz Faisal Bashir, Mohammad Al Arab, Abeer Al Otaibi, Sindiyan Al Shaikh Mubarak, Ali Ahmed Al Qarni

**Affiliations:** 1King Abdullah International Medical Research Center (KAIMRC), Eastern Region, Al Ahsa 31982, Saudi Arabia; alhussainiar@kaimrc.edu.sa (A.A.H.); bashirsha@kaimrc.edu.sa (S.F.B.); alarabmo84@gmail.com (M.A.A.); otaibiabe@kaimrc.edu.sa (A.A.O.); alshaikhmubaraks@kaimrc.edu.sa (S.A.S.M.); qarniaa@mngha.med.sa (A.A.A.Q.); 2Division of Biomedical Research Core Facility, King Saud bin Abdulaziz University for Health Sciences (KSAU-HS), Al Ahsa 36428, Saudi Arabia; 3Ministry of National Guard-Health Affairs (MNG-HA), King Abdulaziz Hospital, Al Ahsa 36428, Saudi Arabia; solimana@mngha.med.sa (A.F.S.); alsubaieema@mngha.med.sa (M.A.S.); khamisko@mngha.med.sa (K.K.O.)

**Keywords:** diabetes, dyslipidemia, insulin resistance, obesity, cytokines, adipokines, cystatin C, oxidized LDL, vascular complications, atherosclerosis, endothelial dysfunction, inflammation

## Abstract

In individuals with type 2 diabetes mellitus (T2DM), elevated levels of both plasma and urinary cystatin C (Cys-C) contribute to increased oxidation, which in turn accelerates the oxidation of low-density lipoprotein (LDL). This process may worsen the development of atherosclerosis and cardiovascular disease by promoting endothelial dysfunction and inflammation. Despite its potential significance, the relationship between Cys-C and oxidized LDL (ox-LDL) in T2DM remains poorly understood. This study investigated the relationship between plasma and urinary Cys-C and ox-LDL levels in T2DM patients. The cohort included 57 patients with T2DM (mean age 61.14 ± 9.99 years; HbA1c 8.66 ± 1.60% and BMI 35.15 ± 6.65 kg/m^2^). Notably, 95% of the patients had hypertension, 82% had dyslipidemia, 59% had an estimated glomerular filtration rate (eGFR) < 60 mL/min/1.73 m^2^, 14% had coronary artery disease (CAD), and 5% had a history of stroke. Plasma and urinary Cys-C and ox-LDL levels were measured using ELISA. Adipokine and cytokine levels were measured using the multiplex^®^ MAP Human Adipokine Magnetic Bead Panels. Spearman’s correlation analysis revealed a significant positive correlation of plasma and urinary Cys-C with ox-LDL (r = 0.569, *p* = 0.0001 and r = 0.485, *p* = 0.0001, respectively). Multivariable regression analysis indicated that both plasma and urinary Cys-C were independently associated with ox-LDL, after adjusting for confounding factors (β = 0.057, *p* = 0.0001 and β = 0.486, *p* = 0.003, respectively). Stepwise linear regression identified TNFα and adiponectin as the strongest predictors of the relationship between urinary Cys-C and ox-LDL (β = 0.382, *p* = 0.0001; r^2^ = 0.64), while adiponectin alone was the best predictor of the plasma Cys-C and ox-LDL association (β = 0.051, *p* = 0.005; r^2^ = 0.46). Furthermore, adiponectin partly mediated the relationship between plasma Cys-C and ox-LDL, explaining 18% of the variance in this association. In contrast, TNFα partly mediated the relationship between urinary Cys-C and ox-LDL, accounting for 28% of the variance. This study emphasizes the complex interaction between Cys-C and ox-LDL in T2DM. It highlights the need for additional research involving larger patient cohorts to improve our understanding of the therapeutic potential of plasma and urinary Cys-C in conjunction with ox-LDL for managing complications associated with T2DM.

## 1. Introduction

Type 2 diabetes mellitus (T2DM), insulin resistance, and dyslipidemia are interrelated in a complex network that exacerbates oxidative stress [1]. Increased oxidative stress leads to endothelial dysfunction, inflammation, and tissue damage, contributing to kidney dysfunction and cardiovascular complications such as atherosclerosis. T2DM-induced hyperglycemia leads to the accumulation of advanced glycation end products (AGEs), causing kidney damage by promoting fibrosis, glomerular hypertension, and hyperfiltration, which impair kidney function over time [2].

Kidney diseases, particularly diabetic kidney disease (DKD), are common complications of T2DM [3]. Persistent high blood sugar levels in T2DM lead to glomerular hyperfiltration, oxidative stress, and inflammation, progressively damaging the renal microvasculature while promoting increased vascular calcification and accelerating the formation of atherosclerotic plaques [4]. This, in turn, impairs kidney function, resulting in albuminuria and a decreased glomerular filtration rate (GFR). There is a well-established link between DKD and atherosclerosis, as both conditions share significant pathophysiological mechanisms, including endothelial dysfunction, chronic inflammation, and dyslipidemia [5,6,7].

Oxidized low-density lipoprotein (ox-LDL) plays a crucial role in developing atherosclerosis. When LDL particles oxidize, they undergo structural changes that make them more likely to promote atherosclerosis. ox-LDL has been shown to contribute to endothelial dysfunction, stimulate foam cell formation, and worsen inflammatory responses, thereby accelerating the development and accumulation of atherosclerotic plaques in the arteries [8]. In T2DM, increased oxidative stress and altered lipid metabolism create conditions that favor LDL oxidation, increasing the risk of atherosclerosis and cardiovascular events [9]. Determining the exact proportion of ox-LDL relative to total LDLc in circulation poses challenges. This ratio can vary due to oxidative stress, diet, metabolic conditions, and health issues like obesity and T2DM. Although the proportion of ox-LDL in the total circulating LDL is small, it can significantly influence cardiovascular health [10,11]. While most LDL particles remain unoxidized, a portion undergoes oxidative modification, leading to ox-LDL formation. Comprehensive data on this specific ratio are limited, but existing research offers insights into ox-LDL concentrations across different populations. For instance, a study involving children and adolescents found that ox-LDL levels were significantly higher in overweight and obese individuals compared to their normal-weight counterparts [12]. However, the study did not specify the exact percentage of oxidized LDL. Additionally, another study demonstrated that the ox-LDL/LDLc ratio was significantly associated with the severity of coronary atherosclerosis in individuals with T2DM. Furthermore, the ox-LDL/LDLc ratio was linked to systolic cardiac dysfunction in patients undergoing permanent hemodiafiltration therapy [13].

Cystatin C (Cys-C) is a low-molecular-weight protein produced by all nucleated cells and filtered by the kidneys, making it a valuable marker for kidney function [14]. Unlike creatinine, Cys-C levels are less influenced by factors such as muscle mass and diet, offering a more accurate reflection of GFR [15,16]. Moreover, Cys-C is not affected by renal tubular secretion or reabsorption, enhancing its usefulness in assessing kidney function, particularly in patients with abnormal muscle mass, obesity, or those taking medications that alter creatinine metabolism [17,18].

Elevated levels of plasma Cys-C are associated with systemic inflammation and endothelial dysfunction, which drive atherosclerosis [19]. In healthy individuals, Cys-C is almost entirely reabsorbed and broken down by the proximal tubules, resulting in minimal excretion in the urine [15]. However, chronic kidney disease can impair the kidneys’ ability to reabsorb Cys-C, leading to increased urinary excretion. Additionally, elevated levels of plasma Cys-C indicate a decline in kidney filtration, while high levels of urinary Cys-C suggest tubular dysfunction or tubulointerstitial damage. In the early stages of kidney disease, urinary Cys-C may rise before even plasma Cys-C, making it a useful marker for detecting kidney injury before significant changes in GFR occur [20,21]. Monitoring both plasma and urinary Cys-C provides a more nuanced understanding of kidney function and disease progression, especially in conditions like T2DM.

The imbalance of adipokines and cytokines in obesity and diabetes plays a crucial role in the formation of ox-LDL, contributing to vascular inflammation and atherosclerosis. Adipokines are signaling molecules secreted by adipose tissue. In obesity, pro-inflammatory adipokines such as leptin and resistin are often increased, while anti-inflammatory adipokines like adiponectin are decreased [22]. This imbalance leads to a chronic state of low-grade inflammation. Similarly, cytokines such as TNF-α, IL-6, and CRP are elevated in obesity and diabetes, further enhancing oxidative stress and inflammation [23,24,25].

Cys-C is sensitive to inflammatory cytokines like TNFα and IL-6, which are often elevated in T2DM, and is influenced by adipokines such as leptin and adiponectin that regulate inflammation and insulin sensitivity [26]. In T2DM, high levels of Cys-C indicate both kidney dysfunction and increased cardiovascular risk due to their association with systemic inflammation, arterial stiffness, and plaque stability [27,28]. This suggests that plasma Cys-C could serve as an early indicator of cardiovascular risk, potentially signaling subclinical atherosclerosis before prominent cardiovascular events arise. In contrast, urinary Cys-C excretion suggests tubular damage caused by local inflammation or injury in the context of atherosclerotic disease due to impaired blood flow resulting from atherosclerotic changes [20].

While conflicting results remain inconclusive regarding the clinical role of Cys-C as a surrogate cardiovascular marker beyond established classical risk factors [19,29,30,31], the complexity of the relationship between Cys-C and ox-LDL may depend on the disease state and levels of oxidative stress. In fact, Cys-C plays a dual role in its interaction with ox-LDL as it can protect against certain harmful effects of ox-LDL, such as oxidative damage and endothelial dysfunction. Simultaneously, elevated Cys-C levels, often observed in conditions like obesity and diabetes, may indicate an ongoing inflammatory response associated with oxidative stress and atherosclerosis. These observations suggest that levels of Cys-C and ox-LDL could serve as complementary biomarkers for assessing the severity and progression of vascular complications, particularly in individuals with obesity and T2DM. Monitoring these two biomarkers might enhance our understanding of cardiovascular disease progression and help identify diabetic patients at greater risk for cardiovascular complications. Furthermore, the interaction between Cys-C and ox-LDL, in the context of abnormal production of adipokines and cytokines linked to T2DM and obesity, may create a complex network that could influence their interaction. Therefore, this study aims to examine the relationship between plasma and urinary Cys-C and ox-LDL levels in patients with T2DM to better understand the mechanisms connecting these factors. This is the first study to investigate the relationship between plasma and urinary Cys-C and ox-LDL levels in the context of obesity and T2DM. Our research reveals distinct patterns in the association of plasma and urinary Cys-C with ox-LDL. It offers new insights into their potential roles as biomarkers for cardiovascular risk in obesity and T2DM. These findings may pave the way for more targeted diagnostic and therapeutic strategies to manage vascular complications in obesity and T2DM.

## 2. Results

### 2.1. Baseline Characteristics of the Study Subjects with T2DM

The characteristics of the study subjects with T2DM are presented in Table 1. The mean age of patients was 60.96 ± 9.99 years, with females making up 44%. Ninety-five percent (95%) had hypertension, 79% had dyslipidemia, and 14% had CAD. The mean HbA1c level was 8.66% ± 1.60%. Most subjects were overweight, with a mean BMI of 35.15 ± 6.65 kg/m^2^. All subjects were Saudi citizens, with 82% having a family history of diabetes and 58% having a family history of cholesterol issues. The mean plasma and urinary Cys-C and ox-LDL levels, as determined by ELISA, were 1357.58 ± 598.90 ng/mL, 105.30 ± 69.04 ng/mL, and 247.97 ± 67.50 ng/mL, respectively.

### 2.2. Quantification of Plasma and Urinary Cys-C and ox-LDL Levels Among T2DM Patients

The urinary Cys-C values in diabetic patients displayed a skewed distribution, with a median of 105.30 [45.55, 173.15] ng/mL. In contrast, plasma Cys-C and ox-LDL levels followed a normal distribution among diabetic patients, each having a median of 1267.01 [897.90, 1787.96] ng/mL and 237.15 [198.79, 288.45] ng/mL, respectively (Figure 1A–C). Furthermore, a positive correlation was noted between plasma Cys-C and urinary Cys-C levels, and both were positively correlated with ox-LDL levels (Figure 1D–F).

### 2.3. Correlations Between Plasma and Urinary Cys-C and ox-LDL Levels and Clinical Parameters in T2DM Patients

Spearman’s correlation analysis revealed that plasma and urinary Cys-C positively correlated with Ox-LDL (r = 0.569, *p* = 0.0001 and r = 0.485, *p* = 0.0001, respectively; Table 2 and Appendix A). Additionally, plasma Cys-C positively correlated with urinary Cys-C (r = 0.279, *p* = 0.036; Table 2 and Appendix A). There were no significant correlations between plasma and urinary Cys-C and other parameters, including age, gender, hypertension, HbA1c, creatinine, total cholesterol, LDL-c, HDL-c, triglycerides, Hs-CRP, BMI, and nitric oxide (Table 2 and Appendix A). Among all quantified adipokines and cytokines, plasma Cys-C showed a positive correlation with IL-6, leptin, NGF, and TNFα (r = 0.397, *p* = 0.020; r = 0.342, *p* = 0.009; r = 0.300, *p* = 0.023; and r = 0.663, *p* = 0.0001, respectively; Table 3 and Appendix A). In contrast, urinary Cys-C positively correlated with TNFα only (r = 0.333, *p* = 0.011; Table 3 and Appendix A). Furthermore, ox-LDL positively correlated with IL-6 and leptin (r = 0.340, *p* = 0.010; r = 0.353, *p* = 0.007; and r = 0.530, *p* = 0.0001, respectively; Table 3 and Appendix A) and showed an inverse correlation with adiponectin (r = −0.465, *p* = 0.0001; Table 3 and Appendix A).

### 2.4. Univariate Regression Analysis for the Association of Plasma and Urinary Cys-C with ox-LDL in T2DM Patients

We first conducted a univariate analysis (Models 1 and 2) to examine the association between plasma and urinary Cys-C and Ox-LDL levels, respectively. The results revealed a positive association between plasma Cys-C, urinary Cys-C, and ox-LDL (β = 0.071, *p* = 0.0001, r^2^ = 0.393; β = 0.443, *p* = 0.0001, r^2^ = 0.205, respectively; Table 4). The association with ox-LDL was more significant for plasma Cys-C than urinary Cys-C (Model 3, r^2^ = 0.680).

### 2.5. Multivariate Regression Analysis for the Association Between Plasma and Urinary Cys-C with ox-LDL in T2DM Patients

We also explored the potential role of additional variables that might explain the relationship between plasma and urinary Cys-C and ox-LDL concentrations. A multiple regression analysis was conducted after adjusting the models for various independent variables such as age, gender, HbA1c, total cholesterol, LDL-c, HDL-c, triglycerides, Hs-CRP, BMI, nitric oxide, CAD, stroke, dyslipidemia, adipokines, and cytokines. The analysis indicated that plasma and urinary Cys-C were independently associated with ox-LDL, even after controlling for potential covariates (β = 0.057, *p* = 0.0001, r^2^ = 0.811, Model M3, Table 5; and β = 0.486, *p* = 0.003, r^2^ = 0.563, Model M2; Table 6). However, these associations diminished when additional independent variables such as adipokines and cytokines were included in the regression models (Model M4, Table 5 and Table 6).

### 2.6. Stepwise Regression Analysis for the Identification of the Best Predictors for the Association with ox-LDL Among Circulating Adipokines and Cytokines in T2DM Patients

To further evaluate the influence of various circulating adipokines and cytokines on the relationship between plasma and urinary Cys-C and ox-LDL, we conducted a stepwise linear regression to identify the optimal subsets of potential predictors with estimated coefficients that best predict the relationship between Cys-C and ox-LDL (Table 7). Independent variables from the selected adipokines and cytokines were added to the regression models with either plasma or urinary Cys-C. The results revealed that adiponectin had the most significant impact when included in the predictive models, explaining 46% of the variability in the relationship between plasma Cys-C and ox-LDL (β = 0.051, *p* = 0.005, r^2^ = 0.461, Model M2; Table 7). In contrast, adiponectin and TNFα together accounted for 64% of the variability in the relationship between urinary Cys-C and ox-LDL (β = 0.382, *p* = 0.001, r^2^ = 0.637, Model M3; Table 7).

### 2.7. Regression Analysis of the Potential Mediating Effects of the Relationship Between Plasma and Urinary Cys-C and ox-LDL in T2DM Patients

To further investigate whether the association between Cys-C (independent variable) and ox-LDL (dependent variable) could be explained by some of the selected adipokines and cytokines (mediators), we conducted sequential regression analyses using the PROCESS macro V4.2 developed by Andrew F. Hayes as described in the methods section. Among all the variables tested, adiponectin demonstrated a partial mediation of the relationship between plasma Cys-C and ox-LDL, with a mediation effect of 18% (Table 8). In contrast, TNFα partially mediated the relationship between urinary Cys-C and ox-LDL, accounting for 28% of the variation in this association (Table 8).

## 3. Discussion

Obesity and T2DM are well-established risk factors for various cardiovascular complications, including endothelial dysfunction and atherosclerosis [32,33]. Cys-C and ox-LDL are important biomarkers involved in the pathophysiology of atherosclerosis, particularly in obesity and T2DM. Cys-C is primarily recognized for its role in regulating cysteine protease activity [34]. It is produced constantly by all nucleated cells and is freely filtered by the glomerulus in the kidneys. Importantly, unlike creatinine, its levels are not significantly affected by age, muscle mass, or diet, making it a more reliable marker of kidney function [35]. Both plasma and urinary Cys-C provide valuable insights into kidney health. Plasma Cys-C is primarily used to assess glomerular filtration, while urinary Cys-C provides information about renal tubular function and injury. Studies have shown that both Cys-C and ox-LDL play important roles during the development of atherosclerosis by promoting endothelial dysfunction and inflammation [19,36,37,38]. In individuals with obesity and T2DM, the combined effects of hyperglycemia, dyslipidemia, and inflammation lead to increased ox-LDL levels, further worsening endothelial dysfunction and increasing the risk of cardiovascular events [39]. Nevertheless, the connection between Cys-C and ox-LDL levels in T2DM remains poorly understood. Understanding the link between Cys-C and ox-LDL may uncover common pathogenic pathways connecting renal dysfunction, oxidative stress, and cardiovascular disease in T2DM. This knowledge could enhance risk stratification, facilitate early complication detection, and support the development of targeted interventions to reduce cardiovascular risk in patients with T2DM.

Previous reports indicated significant increases in Cys-C and ox-LDL levels in cardiovascular and metabolic diseases, including diabetes and obesity. Compared to the published reports on the established value reference ranges in a healthy population (ox-LDL healthy range: <50 ng/mL and Cys-C healthy range: 0.4–1.0 mg/L), our study revealed that circulating levels of ox-LDL (mean value: 247.97 ± 67.50 ng/mL) and Cys-C (mean value: 1357.58 ± 598.90 pg/mL) were significantly elevated among our T2DM subjects, which aligns with previously published studies [14,40,41,42,43]. Notably, there was a positive correlation between plasma Cys-C and urinary Cys-C, and both markers were positively correlated with ox-LDL levels. Although studies have reported significant correlations between Cys-C and ox-LDL with various clinical and biochemical parameters [39,40,44,45,46], we found no significant correlations between Cys-C, ox-LDL, and traditional lipid parameters such as total cholesterol, LDL-c, HDL-c, triglycerides, HbA1c, nitric oxide, BMI, creatinine, and eGFR. The lack of significance with both traditional and non-traditional risk factors may partly be attributed to the size of the population and/or the duration and stage of the disease itself.

Adipokines and cytokines play a significant role in regulating adipose tissue inflammation and metabolic dysfunction in diabetic patients, which can influence levels of both Cys-C and ox-LDL [22]. Increased pro-inflammatory cytokines, such as TNFα, IL-6, and adiponectin, have been shown to contribute to endothelial dysfunction, oxidative stress, and insulin resistance [47,48]. Simultaneously, the upregulation of inflammatory markers promotes the oxidation of LDL particles, resulting in elevated ox-LDL levels [49]. This cascade of events worsens vascular damage and further increases the risk of cardiovascular complications in diabetic individuals [50,51,52,53]. We found that plasma Cys-C exhibited a negative correlation with adiponectin and a positive correlation with IL-6, TNFα, leptin, and NGF. In contrast, urinary Cys-C was negatively correlated with TNFα. These findings align with previous studies, suggesting that variations in adipokine and inflammatory cytokine levels can influence both plasma and urinary Cys-C [54,55,56,57,58]. These results further support that an imbalance of adipokines and inflammatory markers may significantly impact cardiovascular risk factors in individuals with diabetes [52,59,60,61,62,63]. Notably, adiponectin, which is generally considered protective, negatively correlates with ox-LDL, whereas IL-6 promotes LDL oxidation, thereby contributing to oxidative stress and vascular complications [52,59]. Understanding the interplay among all these factors is essential, as changes in adipokine and inflammatory cytokine levels can significantly influence levels of Cys-C and ox-LDL during the course of the disease. We hypothesize that variation in circulating adipokines and inflammatory cytokines may directly or indirectly affect the relationship between Cys-C and ox-LDL, thereby contributing to vascular dysfunction and increasing cardiovascular risk.

Cys-C may also affect lipid metabolism by modulating the release of adipokines from adipose tissue, resulting in changes to the composition of lipoprotein particles [64]. Given that ox-LDL is closely associated with dyslipidemia and atherosclerosis, Cys-C could indirectly affect lipoprotein lipase activity [65] or other enzymes that modify LDL particles, thereby influencing the balance between native and oxidized forms of LDL.

In this study, TNFα and adiponectin emerged as significant confounding factors influencing the relationship between Cys-C and ox-LDL. Notably, adiponectin was found to partially mediate the relationship between plasma Cys-C and ox-LDL, accounting for an 18% mediation effect in this association. In contrast, TNFα partially mediated the relationship between urinary Cys-C and ox-LDL, explaining 28% of the variation in this association. Although the magnitude of these mediating effects is relatively modest, these findings emphasize the distinct roles of adiponectin and TNFα in differentially mediating the interaction between Cys-C and ox-LDL. This interaction may also involve additional, unmeasured confounders not included in the analysis, which could indirectly influence the mediation of the relationship between Cys-C and ox-LDL. The imbalance in adipokines is marked by increased levels of pro-inflammatory adipokines leptin and resistin and decreased levels of anti-inflammatory adiponectin, exacerbating the inflammation process. These adipokines interact with cytokines, including TNF-α, IL-6, and IL-1β, which play a role in the inflammatory response and impair insulin signaling pathways [66]. In fact, it is well-established that the interaction between adiponectin and TNFα is crucial to the development and progression of T2DM [67]. Reduced adiponectin levels promote increased TNFα production, while elevated TNFα levels inhibit adiponectin secretion, creating a feedback loop that worsens insulin resistance and systemic inflammation, ultimately leading to late renal injury resulting in glomerular damage, fibrosis, and eventual kidney failure [68]. Additionally, ox-LDL amplifies systemic inflammation, creating a feedback loop in which oxidized LDL exacerbates the inflammatory environment driven by cytokines and adipokines. In contrast, elevated inflammatory cytokines foster further oxidation of LDL [37].

We have demonstrated that adiponectin significantly mediates the relationship between plasma Cys-C and ox-LDL, suggesting it may influence systemic inflammation and lipid metabolism. As an anti-inflammatory factor that enhances insulin sensitivity, adiponectin could also help modulate the oxidative modification of LDL in circulation, potentially reducing the risk of atherosclerosis [69]. In contrast, pro-inflammatory cytokine TNFα mediates the relationship between urinary Cys-C and ox-LDL, indicating that the renal system, possibly through inflammation, is more closely regulated by TNFα, potentially contributing to renal and cardiovascular diseases [70].

The results of the current study may not be sufficient to identify a potential mechanism for the observed mediating effect, while moderate, on the association between Cys-C and ox-LDL in obese individuals with T2DM. One plausible explanation is that the mediating effect of adiponectin on the relationship between plasma Cys-C and ox-LDL may depend on its production levels. For instance, when adiponectin levels are normal or elevated, it can suppress the inflammatory pathways leading to LDL oxidation and lower Cys-C levels [71]. This would provide a protective effect by reducing both the oxidative modification of LDL and the renal stress associated with the high Cys-C levels observed in inflammatory diseases. Conversely, if adiponectin levels are low, as seen in obesity, this could exacerbate the harmful effects of ox-LDL [72], thereby increasing Cys-C levels and heightening the risk of cardiovascular disease. In contrast, TNF-α can significantly influence the relationship between urinary Cys-C and ox-LDL due to its strong inflammatory and pro-oxidative effects. When TNF-α levels rise, as in chronic inflammatory and metabolic diseases, urinary Cys-C and ox-LDL levels typically increase, indicating underlying kidney dysfunction and oxidative stress.

The observed differential roles of adiponectin and TNFα suggest that plasma and urinary Cys-C reflect distinct aspects of metabolism and inflammation. Plasma Cys-C may be related to systemic metabolic or inflammatory states, while urinary cystatin appears more connected to kidney-specific inflammation, with TNFα playing a central role. These findings highlight the complex interactions between inflammation, lipid metabolism, and organ-specific factors. Furthermore, the interaction between plasma and urinary Cys-C and ox-LDL in individuals with obesity and T2DM may synergistically affect inflammation and endothelial dysfunction, contributing to atherosclerosis. Cys-C might modulate the expression of adhesion molecules and cytokines that promote the infiltration of inflammatory cells into the arterial wall. This inflammatory environment promotes the oxidation of LDL particles, leading to increased ox-LDL formation. Additionally, ox-LDL exacerbates endothelial injury and facilitates its uptake into endothelial cells, smooth muscle cells, and macrophages, forming foam cells and progressing atherosclerotic plaques [73]. Elevated levels of Cys-C in plasma may indicate increased activation of these pathways, including the upregulation of matrix metalloproteinases (MMPs), which play a role in vascular remodeling [74]. This process creates a feedback loop that accelerates atherosclerotic plaque formation, destabilization, and potential rupture. Additionally, in T2DM, elevated oxidative stress increases reactive oxygen species (ROS), which can lead to intensified oxidation and, consequently, higher levels of ox-LDL [75]. This process can influence Cys-C levels and lead to kidney dysfunction [76]. Furthermore, endothelial dysfunction can further reduce renal blood flow, contributing to kidney injury and disrupting the glomerular filtration barrier [76]. This disruption may release Cys-C into the urine as an early kidney injury biomarker. Our findings indicate that using Cys-C and ox-LDL as dual biomarkers for cardiovascular risk stratification in T2DM could offer a more comprehensive risk evaluation than relying on each marker individually. By examining their relationship, a synergistic predictive model for cardiovascular and renal complications could be developed, allowing for improved monitoring and personalized treatment strategies.

One of the main limitations of this study is its relatively small sample size and the lack of a healthy control group, which may limit the statistical power and generalizability of these findings. A larger cohort would help confirm the observed association between Cys-C and ox-LDL and lead to more robust conclusions. Another potential limitation is the inability to account for all possible confounding factors (e.g., medication, including lipid-lowering drugs and other missing adipokines and inflammatory cytokines), as there may be undetected variables that could directly or indirectly influence the study’s outcome. Furthermore, this study specifically focused on obese individuals with T2DM, making the findings potentially less applicable to other cohorts and different stages of the disease. Additional studies with larger sample sizes, more diverse populations, and a wider range of potential confounders are needed to validate these findings.

## 4. Materials and Methods

### 4.1. Study Population and Protocol

This retrospective study involved 57 patients with T2DM recruited from clinics at King Abdulaziz Hospital (KAH), part of the Ministry of National Guard-Health Affairs in Al-Ahsa, Kingdom of Saudi Arabia, between January 2020 and April 2021. The study protocol received approval from the Institutional Review Board of the Ministry of National Guard, Health Affairs (IRB protocol# IRBC/1972/18), and written informed consent was obtained from each participant. Participants were excluded from the study if, at baseline, they met one or more of the following criteria: being on chronic renal replacement therapy (such as hemodialysis, peritoneal dialysis, or transplantation); having a history of active malignancy (excluding basal cell carcinoma) within the last five years (or prostatic cancer within the past two years); systemic lupus erythematosus and other autoimmune diseases affecting kidney function; a history of type 1 diabetes, acute infection or fever, pregnancy, chronic viral hepatitis, or HIV infection; or current unstable cardiac disease.

### 4.2. Measurement of Plasma Levels of Cys-C, Ox-LDL, Adipokines, and Cytokines in T2DM Patients

Fasting blood samples were collected in the morning after a minimum 12 h fast, placed into EDTA-containing tubes, and centrifuged at 4 °C at 3000 rpm for 10 min to separate the plasma for biochemical tests. Additionally, urine specimens were collected and stored on ice for transportation. Within 2 h of collection, all samples were clarified by centrifugation at 3000 rpm for 10 min. Plasma and urine samples from T2DM patients were aliquoted and stored at −80 °C until further analysis. Patient medical history, demographics, and laboratory parameters were extracted from the electronic medical record in the BEST Care database. Concentrations of plasma and urinary Cys-C and ox-LDL were determined by ELISA, using commercially available kits according to the manufacturer’s protocols (catalog# E-EL-H3643 and catalog# E-EL-H6021, respectively, from Elabscience Biotechnology Inc., Houston, TX, USA). Adipokine and cytokine levels were measured using the multiplex^®^ MAP Human Adipokine Magnetic Bead Panel 2 (catalog# HADK2MAG-61K; EMD Millipore-Sigma; Burlington, MA, USA). Human adiponectin/ACRP30 concentrations were assessed using a commercially available ELISA kit (catalog# E-EL-H6122; Elabscience Biotechnology Inc., Houston, TX, USA). According to the manufacturer, the ox-LDL kit has a sensitivity of 37.5 pg/mL, with a detection range of 62.5 to 4000 pg/mL. In contrast, Cys-C has a sensitivity of 0.19 ng/mL, with a detection range of 0.31 to 20 ng/mL. These ELISA kits specifically identify human Cys-C and ox-LDL in samples such as serum, plasma, and other biological fluids, showing no significant cross-reactivity. The plasma and urinary concentrations of samples were determined in duplicates.

### 4.3. Statistical Analysis

Statistical analyses were performed using SPSS software version 30.0 (IBM Corp., Chicago, IL, USA). The normality of the data was evaluated using a Kolmogorov–Smirnov test. Continuous variables that showed a normal distribution were presented as the mean ± SD, while categorical variables were reported as frequencies and percentages. We utilized a non-parametric Spearman’s correlation test to assess the correlation between Cys-C, ox-LDL levels, and other variables. We conducted a multiple regression analysis to investigate the association between Cys-C and ox-LDL before and after adjusting the model for variables such as age, sex, HbA1c, creatinine, BMI, total cholesterol, LDL-c, HDL-c, triglycerides, and selected adipokines and cytokines. A stepwise analysis was performed to identify the optimal subset of predictor variables influencing the association between Cys-C and ox-LDL. Regression analysis results were reported as β-coefficients and 95% confidence intervals. For statistical significance, two-sided tests with *p*-values < 0.05 were considered significant. To determine how much the measured effect of the independent variable (Cys-C) on the dependent variable (ox-LDL) could be attributed to a potential mediator variable (adipokine or cytokine), we evaluated the mediation effect in sequential regression analyses using the PROCESS macro V4.2 software developed by Andrew F. Hayes (Model 4). The path model simultaneously tested three effects for each mediator: (i) the effect of the independent variable (Cys-C) on the mediator (indirect effect path a); (ii) the effect of the mediator on the dependent variable (ox-LDL), known as the indirect effect path b; and (iii) the mediation effect (a*b), reflecting the reduction in the relationship between Cys-C and ox-LDL (total effect path c) when the mediator is included in the model (direct effect path c’). We also applied bootstrapping methods to calculate the 95% confidence interval (CI) of coefficients for the total, indirect, and direct effects. The mediation outcome is statistically significant if the CI does not include zero. Partial mediation (effect < 100%) was observed when the mediating variable significantly diminished the association between Cys-C and ox-LDL. The mediation effect percentage was defined as the ratio of the indirect effect to the total effect.

## 5. Conclusions

In conclusion, our results clearly demonstrate distinct patterns in the interaction of plasma and urinary Cys-C with ox-LDL, establishing their significance as biomarkers for assessing cardiovascular risk in individuals with obesity and T2DM. Additionally, we have shown that adiponectin and TNFα differentially mediate the interaction of plasma and urinary Cys-C with ox-LDL, respectively (Figure 2). These mediating effects may partially explain the relationship between Cys-C and ox-LDL. However, other factors or additional mediators could also play a role in this association. The relatively moderating effect indicates that while adiponectin and TNFα influence the relationship between Cys-C and ox-LDL, they do not fully account for it. Further research is essential to understand the mechanisms linking Cys-C changes with ox-LDL and clarify their causal relationship and clinical significance in managing vascular complications associated with obesity and T2DM. In such studies, clinicians can more accurately identify high-risk individuals by monitoring plasma and urinary Cys-C levels in conjunction with ox-LDL. This could facilitate the implementation of targeted and personalized strategies to manage cardiovascular risk, prevent the progression of atherosclerosis, and improve outcomes for diabetic patients.

## Figures and Tables

**Figure 1 ijms-26-03001-f001:**
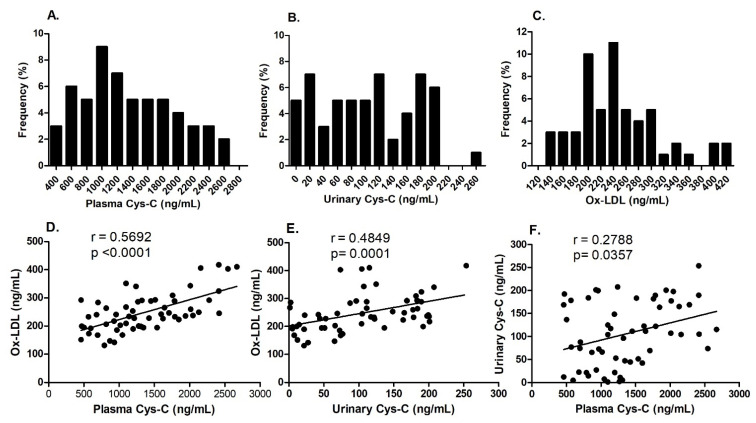
Plasma Cys-C, urinary Cys-C, and ox-LDL levels were measured using ELISA in patients with T2DM. The plasma Cys-C, urinary Cys-C, and ox-LDL distribution frequencies were calculated for the study cohort (**A**–**C**). The correlations between plasma Cys-C, urinary Cys-C, and ox-LDL were assessed using Spearman’s correlation analysis (**D**–**F**).

**Figure 2 ijms-26-03001-f002:**
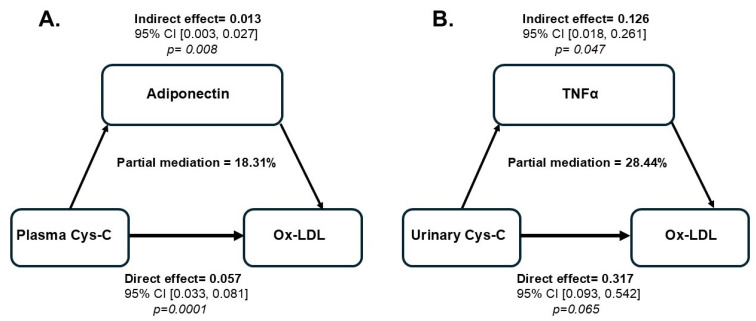
Estimation of the proportion of associations between plasma and urinary Cys-C and ox-LDL mediated by adiponectin (**A**) or TNF-α (**B**). Each moderator (adiponectin or TNF-α) was entered individually into the regression model using the PROCESS macro software developed by Andrew F. Hayes (Model 4). The bootstrapping method estimated the confidence interval (CI) of the mediating effect. A *p*-value of ≤0.05 is considered significant.

**Table 1 ijms-26-03001-t001:** Baseline characteristics of the 57 diabetic participants.

Baseline Characteristics	T2DM Cohort (N = 57)
Age (years) *	60.96 ± 9.99
BMI (kg/m^2^) *	35.15 ± 6.65
Normal Weight: BMI 18.5–24.9; N (%)	3 (5.26)
Overweight: BMI 25–29.9; N (%)	8 (14.03)
Obesity Class I (Moderate): BMI 30–34.9; N (%)	21 (36.84)
Obesity Class II (Severe): BMI 35–39.9; N (%)	13 (22.81)
Obesity Class III (Very Severe or Morbidity Obese): BMI ≥ 40; N (%)	12 (21.05)
Systolic BP (mmHg)	145.00 (123.50, 153.50)
Diastolic BP (mmHg)	71.00 (61.50, 81.00)
HbA1c (%) *	8.66 ± 1.60
Fasting Glucose (mg/dL)	8.00 (6.67, 14.27)
Hs-CRP (mg/L)	7.60 (3.40, 25.97)
Creatinine (µmol/L)	111.50 (81.00, 142.25)
eGFR (mL/min/1.73 m^2^) *	58.04 ± 26.50
Total Cholesterol (mg/dL)	154.68 (116.01, 193.35)
LDL-c (mg/dL)	77.34 (77.34, 116.01)
HDL-c (mg/dL)	38.67 (38.67, 38.67)
Triglycerides (mg/dL)	177.14 (88.57, 177.14)
ApoB (mg/mL)	86.00 (78.00, 103.50)
Pl-Cys-C (ng/mL) *	1357.58 ± 598.90
Ur-Cys-C (ng/mL)	104.96 (111.33, 143.33)
Ox-LDL (ng/mL) *	247.97 ± 67.50
Ox-LDL/LDLc (mg/dL)	251.75 (174.90, 317.95)
Ox-LDL/LDLc (%)	0.025 (0.017, 0.034)
NO (µmol/L) *	130.27 ± 3.01
Adiponectin (pg/mL) *	141.87 ± 63.22
NGF (pg/mL)	8.80 (6.99, 11.60)
IL8 (pg/mL)	13.98 (9.75, 18.45)
MCP1 (pg/mL) *	233.46 ± 93.92
IL1b (pg/mL)	0.93 (0.60, 1.60)
GLP1 (pg/mL)	562.06 (418.74, 857.79)
IL6 (pg/mL)	24.72 (16.25, 32.93)
Insulin (pg/mL)	1615.57 (1180.50, 2386.68)
Leptin (pg/mL) *	8593.25 ± 3556.26
TNFα (pg/mL)	38.80 (32.01, 52.55)
**Family Disease History**	
Diabetes (%)	81.80
Hypertension (%)	25.50
CAD (%)	38.20
Cholesterol (%)	57.70
Stroke (%)	14.30
**Medication**	
Insulin (%)	47.37
HMG-CoA reductase inhibitors (%)	19.30
Metformin (%)	14.03
DPP4 inhibitors (%)	14.03
Sulfonylurea (%)	10.53
Calcium channel blockers (%)	8.77
ACE inhibitors (%)	7.02
NSAID (%)HMG-CoA reductase inhibitors (%)	7.02
Diuretics (%)	3.51
PPI (%)	3.51

Data are presented for continuous variables as the mean (standard deviation, SD) or median (interquartile range, IQR), and as frequencies (percentages, %) for categorical variables. *, values are normally distributed. Abbreviations: BMI, body mass index; BP, blood pressure; HbA1c, hemoglobin A1c; LDL-c, low-density lipoprotein cholesterol; HDL-c, high-density lipoprotein cholesterol; Hs-CRP, high-sensitivity C-reactive protein; eGFR, estimated glomerular filtration rate; ApoB, apolipoprotein B; PlCys-C, plasma cystatin C; UrCys-C, urinary cystatin C; Ox-LDL, oxidized low-density lipoprotein; NO, nitric oxide; NGF, nerve growth factor; IL-8, interleukin 8; MCP-1, monocyte chemoattractant protein-1; IL-1β, interleukin 1 beta; GLP-1, glucagon-like peptide 1; IL-6, interleukin 6; TNFα, tumor necrosis factor alpha. CAD, coronary artery disease; DPP4, dipeptidyl-peptidase 4; ACE, angiotensin-converting-enzyme; NSAID, non-steroidal anti-inflammatory drugs; PPI, proton-pump inhibitor.

**Table 2 ijms-26-03001-t002:** Heatmap Spearman’s correlation coefficient matrix of plasma and urinary cystatin C and ox-LDL with other variables in T2DM patients.

Variables	PlCys-C	UrCys-C	Age	Gender	HbA1c	Cre	Chol-t	LDL-c	HDL-c	TAG	Hs-CRP	BMI	Ox-LDL	NO	eGFR
PlCys-C	1	0.279	0.065	−0.185	−0.062	0.215	−0.089	−0.121	−0.033	0.124	−0.046	−0.172	0.569	0.042	−0.242
UrCys-C	0.279	1	−0.007	0.055	−0.058	−0.071	0.139	0.05	−0.182	0.031	−0.03	0.043	0.485	−0.181	−0.077
Age	0.065	−0.007	1	−0.076	0.044	0.186	−0.191	−0.241	0.042	−0.047	0.013	−0.063	0.082	−0.171	−0.384
Gender	−0.185	0.055	−0.076	1	−0.023	−0.429	0.116	0.36	−0.06	−0.292	0.053	0.167	0.064	−0.185	0.228
HbA1c	−0.062	−0.058	0.044	−0.023	1	−0.003	0.062	0.1	0.178	0.063	0.147	0.204	−0.098	0.079	0.028
Cre	0.215	−0.071	0.186	−0.429	−0.003	1	−0.089	−0.113	−0.134	0.091	0.083	−0.166	0.099	0.247	−0.864
Chol-t	−0.089	0.139	−0.191	0.116	0.062	−0.089	1	0.661	−0.253	0.297	0.056	0.091	−0.048	0.04	0.165
LDL-c	−0.121	0.05	−0.241	0.36	0.1	−0.113	0.661	1	−0.107	0.083	0.081	0.061	−0.032	0.002	0.173
HDL-c	−0.033	−0.182	0.042	−0.06	0.178	−0.134	−0.253	−0.107	1	0.064	0.219	−0.236	−0.079	−0.144	0.165
TAG	0.124	0.031	−0.047	−0.292	0.063	0.091	0.297	0.083	0.064	1	0.165	0.109	−0.206	−0.085	0.011
Hs-CRP	−0.046	−0.03	0.013	0.053	0.147	0.083	0.056	0.081	0.219	0.165	1	0.26	−0.149	−0.123	−0.141
BMI	−0.172	0.043	−0.063	0.167	0.204	−0.166	0.091	0.061	−0.236	0.109	0.26	1	−0.119	0.002	0.107
Ox-LDL	0.569	0.485	0.082	0.064	−0.098	0.099	−0.048	−0.032	−0.079	−0.206	−0.149	−0.119	1	0.152	−0.241
NO	0.042	−0.181	−0.171	−0.185	0.079	0.247	0.04	0.002	−0.144	−0.085	−0.123	0.002	0.152	1	−0.123
eGFR	−0.242	−0.077	−0.384	0.228	0.028	−0.864	0.165	0.173	0.165	0.011	−0.141	0.107	−0.241	−0.123	1
		Heatmap scale	1.000	0.750	0.500	0.250	0.000	−0.250	−0.500	−0.750	−1.000				

**Table 3 ijms-26-03001-t003:** Heatmap Spearman’s correlation coefficient matrix of plasma and urinary cystatin C and ox-LDL with metabolic hormones and cytokines in T2DM patients.

Variables	PlCys-C	UrCys-C	Ox-LDL	ADP	NGF	IL−8	MCP1	IL-1b	GLP1	IL-6	INS	LEP	TNFα
PlCys-C	1	0.279	0.569	−0.346	0.3	0.024	0.104	0.051	0.194	0.397	0.04	0.342	0.663
UrCys-C	0.279	1	0.485	−0.182	0.031	−0.084	−0.04	−0.203	0.214	0.164	0.177	0.223	0.333
Ox-LDL	0.569	0.485	1	−0.465	0.175	0.07	0.066	−0.136	0.225	0.34	−0.148	0.353	0.53
ADP	−0.346	−0.182	−0.465	1	−0.021	0.128	0.152	−0.109	−0.143	−0.068	0.047	−0.179	−0.276
NGF	0.3	0.031	0.175	−0.021	1	0.38	0.027	0.355	0.115	0.348	0.109	0.283	0.375
IL-8	0.024	−0.084	0.07	0.128	0.38	1	0.149	0.427	−0.061	0.265	−0.072	−0.181	0.364
MCP1	0.104	−0.04	0.066	0.152	0.027	0.149	1	−0.098	0.095	0.183	−0.038	0.102	0.171
IL-1b	0.051	−0.203	−0.136	−0.109	0.355	0.427	−0.098	1	−0.207	0.089	−0.034	−0.185	0.153
GLP1	0.194	0.214	0.225	−0.143	0.115	−0.061	0.095	−0.207	1	0.008	0.316	0.053	0.114
IL-6	0.397	0.164	0.34	−0.068	0.348	0.265	0.183	0.089	0.008	1	0.03	0.353	0.529
INS	0.04	0.177	−0.148	0.047	0.109	−0.072	−0.038	−0.034	0.316	0.03	1	0.204	0.005
LEP	0.342	0.223	0.353	−0.179	0.283	−0.181	0.102	−0.185	0.053	0.353	0.204	1	0.237
TNFα	0.663	0.333	0.53	−0.276	0.375	0.364	0.171	0.153	0.114	0.529	0.005	0.237	1
		Heatmap scale	1.000	0.750	0.500	0.250	0.000	−0.250	−0.500	−0.750	−1.000		

Results are expressed as Spearman’s Rho coefficient (R) for 2-tailed Spearman’s correlation analysis. Abbreviations: ox-LDL, oxidized low-density lipoprotein; ADP, adiponectin; NGF, nerve growth factor; IL-8, interleukin 8; MCP1, monocyte chemoattractant protein-1; IL-1b, interleukin 1b; GLP1, glucagon-like peptide 1; IL-6, interleukin 6; INS, insulin; LEP, leptin; TNFα, tumor necrosis factor alpha; Hs-CRP, high-sensitivity C-reactive protein.

**Table 4 ijms-26-03001-t004:** Regression analysis between plasma and urinary cystatin C and ox-LDL in T2DM patients.

Model 1 (R^2^ = 0.393) DV: Ox-LDL	Unstandardized Coefficients		Standardized Coefficients	t	P	95% CI for B	
	B	SE	B			Lower Bound	Upper Bound
(Constant)	152.000	17.539		8.666	0.0001	116.851	187.150
PlCys-C	0.071	0.012	0.627	5.972	**0.0001**	0.047	0.094
Model 2 (R^2^ = 0.205) DV: Ox-LDL	Unstandardized Coefficients		Standardized Coefficients	t	P	95% CI for B	
	B	SE	B			Lower Bound	Upper Bound
(Constant)	201.300	14.759		13.640	0.0001	171.723	230.877
UrCys-C	0.443	0.118	0.453	3.771	**0.0001**	0.208	0.679
Model 3 (R^2^ = 0.680) DV: Ox-LDL	Unstandardized Coefficients		Standardized Coefficients	t	P	95% CI for B	
	B	SE	B			Lower Bound	Upper Bound
(Constant)	137.146	17.576		7.803	0.0001	101.909	172.383
PlCys-C	0.060	0.012	0.537	5.088	**0.0001**	0.037	0.084
UrCys-C	0.273	0.103	0.279	2.645	**0.011**	0.066	0.479

Univariate regression analysis was performed by including PlCys-C in Model 1 and UrCys-C in Model 2. Following this, both PlCys-C and UrCys-C were included in Model 3. P represents the probability value for the regression analysis, with a *p* value ≤ 0.05 being considered significant. CI denotes the confidence interval for the beta (B) coefficient. The predictors (independent variables) are PlCys-C and UrCys-C, while the dependent variable (DV) is ox-LDL. R^2^ indicates the proportion of variance in the dependent variable explained by the independent variables in the linear regression models.

**Table 5 ijms-26-03001-t005:** Multivariate regression analysis for the association of plasma Cys-C with ox-LDL in T2DM patients.

Model 1 (R^2^ = 0.383) DV: Ox-LDL		Unstandardized Coefficients		P	95% CI for B	
		B	SE		Lower Bound	Upper Bound
M1	(Constant)	127.439	65.707	0.058	−4.538	259.416
	PlCys-C	0.067	0.013	**0.0001**	0.042	0.092
	Age	0.577	0.724	0.429	−0.877	2.030
	Gender	11.137	14.578	0.448	−18.144	40.419
	HbA1c	−2.538	4.563	0.580	−11.704	6.627
Model 2 (R^2^ = 0.607) DV: Ox-LDL		Unstandardized Coefficients		P	95% CI for B	
M2	(Constant)	B	SE		Lower Bound	Upper Bound
		60.009	127.676	0.643	−204.108	324.127
	PlCys-C	0.067	0.018	**0.0001**	0.030	0.103
	Age	1.782	1.062	0.107	−0.415	3.979
	Gender	21.245	21.271	0.328	−22.758	65.249
	HbA1c	5.621	6.580	0.402	−7.990	19.232
	Creatinine	−0.001	0.101	0.989	−0.211	0.208
	Cholesterol-t	15.941	14.139	0.271	−13.309	45.190
	LDL-c	−8.344	16.309	0.614	−42.082	25.394
	HDL-c	−5.630	26.820	0.836	−61.111	49.851
	Triglycerides	−19.989	11.079	0.084	−42.907	2.930
	Hs-CRP	−0.367	0.283	0.207	−0.952	0.218
	BMI	−2.707	1.590	0.102	−5.996	0.583
	NO	−0.009	0.417	0.983	−0.871	0.853
Model 3 (R^2^ = 0.811) DV: Ox-LDL		Unstandardized Coefficients		P	95% CI for B	
M3	(Constant)	B	SE		Lower Bound	Upper Bound
		104.326	120.518	0.398	−148.873	357.525
	PlCys-C	0.057	0.014	**0.0001**	0.027	0.086
	Age	1.238	0.962	0.215	−0.784	3.260
	Gender	53.418	21.976	0.026	7.248	99.589
	HbA1c	7.770	5.343	0.163	−3.456	18.996
	Creatinine	0.039	0.091	0.671	−0.152	0.231
	Cholesterol-t	55.064	16.787	0.004	19.795	90.333
	LDL-c	−49.867	18.394	0.014	−88.512	−11.222
	HDL-c	26.252	23.360	0.276	−22.826	75.330
	Triglycerides	−32.158	11.143	0.010	−55.568	−8.748
	Hs-CRP	−0.580	0.235	0.024	−1.074	− 0.087
	BMI	−3.170	1.315	0.027	−5.933	− 0.407
	NO	−0.337	0.339	0.333	−1.049	0.374
	CAD	−23.602	13.669	0.101	−52.319	5.116
	Stroke	−0.124	0.102	0.239	−0.338	0.090
	Dyslipidemia	0.057	0.014	0.699	−60.451	41.412
Model 4 (R^2^ = 0.903) DV: Ox-LDL		Unstandardized Coefficients		P	95% CI for B	
M4	(Constant)	B	SE		Lower Bound	Upper Bound
		50.543	155.125	0.753	−307.175	408.261
	PlCys-C	0.036	0.025	**0.191**	−0.022	0.094
	Age	1.275	1.344	0.371	−1.825	4.375
	Gender	68.092	31.747	0.064	−5.116	141.300
	HbA1c	12.080	7.287	0.136	−4.724	28.884
	Creatinine	0.052	0.126	0.692	−0.238	0.342
	Cholesterol-t	32.151	25.284	0.239	−26.153	90.456
	LDL-c	−21.341	30.869	0.509	−92.524	49.843
	HDL-c	−4.117	31.061	0.898	−75.742	67.509
	Triglycerides	−17.046	15.794	0.312	−53.468	19.376
	Hs-CRP	−0.311	0.316	0.354	−1.038	0.417
	BMI	−4.318	1.860	0.049	−8.608	−0.027
	NO	−0.276	0.624	0.670	−1.716	1.164
	CAD	−5.194	56.769	0.929	−136.104	125.716
	Stroke	−29.095	21.624	0.215	−78.960	20.770
	Dyslipidemia	0.442	41.628	0.992	−95.553	96.436
	Adiponectin	−0.014	0.229	0.953	−0.541	0.514
	NGF	−3.715	4.077	0.389	−13.116	5.687
	IL-8	0.088	2.696	0.975	−6.129	6.306
	MCP1	0.016	0.165	0.925	−0.364	0.395
	IL-1b	−0.129	7.983	0.987	−18.537	18.278
	GLP1	−0.007	0.034	0.842	−0.086	0.072
	IL-6	0.274	0.956	0.782	−1.930	2.479
	Insulin	0.007	0.012	0.591	−0.021	0.035
	Leptin	0.007	0.005	0.196	−0.004	0.018
	TNFα	0.861	1.215	0.499	−1.941	3.664

P represents the probability value for each independent variable in the regression model (M1-3). A *p* value of ≤0.05 is considered significant. CI refers to the confidence interval for the beta (B) coefficient. The predictors (independent variables) include Cys-C and other variables such as age, gender, HbA1C, total cholesterol, LDL-c, HDL-c, triglycerides, Hs-CRP, and BMI. Multiple regression analysis was conducted for the dependent variable (DV): ox-LDL. R^2^ indicates the proportion of variance in the dependent variable (DV) explained by the independent variable in the linear regression model.

**Table 6 ijms-26-03001-t006:** Multivariate regression analysis for the association of urinary Cys-C with ox-LDL in T2DM patients.

Model 1 (R^2^ = 0.258) DV: Ox-LDL		Unstandardized Coefficients		P	95% CI for B	
		B	SE		Lowe Bound	Upper Bound
M1	(Constant)	198.644	68.846	0.006	60.362	336.926
	UrCys-C	0.445	0.114	**0.0001**	0.216	0.675
	Age	0.617	0.794	0.441	−0.978	2.211
	Gender	−4.252	15.750	0.788	−35.886	27.382
	HbA1c	−3.611	4.989	0.473	−13.632	6.410
Model 2 (R^2^ = 0.563) DV: Ox-LDL		Unstandardized Coefficients		P	95% CI for B	
M2	(Constant)	B	SE		Lower Bound	Upper Bound
		108.111	131.725	0.420	−164.384	380.606
	UrCys-C	0.486	0.149	**0.003**	0.178	0.794
	Age	1.498	1.134	0.200	−0.848	3.843
	Gender	18.101	22.374	0.427	−28.182	64.384
	HbA1c	5.069	6.925	0.472	−9.257	19.394
	Creatinine	0.092	0.104	0.384	−0.123	0.308
	Cholesterol-t	0.690	15.143	0.964	−30.636	32.015
	LDL-c	11.360	17.043	0.512	−23.896	46.616
	HDL-c	−22.347	27.726	0.429	−79.701	35.008
	Triglycerides	−8.502	12.033	0.487	−33.394	16.389
	Hs-CRP	−0.346	0.299	0.259	−0.965	0.272
	BMI	−2.496	1.673	0.149	−5.956	0.965
	NO	0.031	0.449	0.945	−0.897	0.960
Model 3 (R^2^ = 0.701) DV: Ox-LDL		Unstandardized Coefficients		P	95% CI for B	
M3	(Constant)	B	SE		Lower Bound	Upper Bound
		214.187	145.383	0.158	−91.252	519.626
	UrCys-C	0.309	0.158	**0.066**	−0.022	0.641
	Age	0.853	1.221	0.494	−1.713	3.419
	Gender	45.786	27.655	0.115	−12.315	103.886
	HbA1c	5.412	6.711	0.430	−8.686	19.511
	Creatinine	0.139	0.111	0.225	−0.094	0.372
	Cholesterol-t	40.844	22.692	0.089	−6.829	88.517
	LDL-c	−29.745	24.665	0.243	−81.565	22.076
	HDL-c	9.456	29.319	0.751	−52.142	71.053
	Triglycerides	−25.015	14.863	0.110	−56.242	6.211
	Hs-CRP	−0.595	0.301	0.064	−1.227	0.037
	BMI	−2.878	1.656	0.099	−6.358	0.601
	NO	−0.392	0.455	0.401	−1.347	0.564
	CAD	−24.604	17.187	0.169	−60.713	11.504
	Stroke	−0.086	0.131	0.519	−0.362	0.189
	Dyslipidemia	−20.048	30.350	0.517	−83.811	43.715
Model 4 (R^2^ = 0.890) DV: Ox-LDL		Unstandardized Coefficients		P	95% CI for B	
M4	(Constant)	B	SE		Lower Bound	Upper Bound
		87.285	161.315	0.603	−284.708	459.278
	UrCys-C	0.149	0.161	**0.382**	−0.222	0.521
	Age	1.390	1.427	0.358	−1.899	4.680
	Gender	55.839	33.417	0.133	−21.222	132.899
	HbA1c	9.570	7.959	0.264	−8.783	27.923
	Creatinine	0.110	0.129	0.420	−0.188	0.408
	Cholesterol-t	21.566	26.966	0.447	−40.616	83.749
	LDL-c	−16.295	33.218	0.637	−92.896	60.306
	HDL-c	−5.630	33.057	0.869	−81.860	70.600
	Triglycerides	−14.535	16.970	0.417	−53.668	24.598
	Hs-CRP	−0.220	0.330	0.523	−0.980	0.540
	BMI	−3.213	1.950	0.138	−7.709	1.283
	NO	0.182	0.652	0.788	−1.322	1.685
	CAD	−25.028	57.626	0.676	−157.914	107.858
	Stroke	−15.093	20.842	0.490	−63.155	32.970
	Dyslipidemia	−19.615	42.016	0.653	−116.504	77.274
	Adiponectin	−0.200	0.216	0.380	−0.697	0.297
	NGF	−1.811	4.212	0.679	−11.523	7.901
	IL-8	−1.296	2.638	0.636	−7.379	4.786
	MCP1	−0.062	0.167	0.720	−0.447	0.323
	IL-1b	−5.192	7.697	0.519	−22.940	12.557
	GLP1	0.004	0.035	0.904	−0.076	0.085
	IL-6	0.635	1.014	0.549	−1.702	2.972
	Insulin	0.000	0.012	0.988	−0.028	0.028
	Leptin	0.004	0.005	0.472	−0.008	0.016
	TNFα	1.649	1.091	0.169	−0.866	4.164

P: the probability value for each independent variable included in the regression models (M1-3). A *p* value ≤ 0.05 is considered significant. CI: confidence interval for the beta (B) coefficient. Predictors (independent variables): Cys-C was included in the regression models alongside other variables such as age, gender, HbA1C, total cholesterol, LDL-c, HDL-c, triglycerides, Hs-CRP, and BMI. Multiple regression analysis was performed for the dependent variable (DV): ox-LDL. R^2^ represents the proportion of variance in the dependent variable (DV) explained by the independent variable in the linear regression model.

**Table 7 ijms-26-03001-t007:** Stepwise regression analysis for the association between plasma and urinary Cys-C with ox-LDL in T2DM patients.

**Plasma Cystatin C Interaction with Ox-LDL**						
Model 1 (R^2^ = 0.369) DV: Ox-LDL		Unstandardized Coefficients		P	95% CI for B	
		B	SE		Lower Bound	Upper Bound
M1	(Constant)	146.402	24.371	0.0001	96.761	196.043
	PlCys-C	0.069	0.016	**0.0001**	0.037	0.102
Model 2 (R^2^ = 0.461) DV: Ox-LDL		Unstandardized Coefficients		P	95% CI for B	
M2	(Constant)	B	SE		Lower Bound	Upper Bound
		225.484	41.238	<0.001	141.378	309.590
	PlCys-C	0.051	0.017	**0.005**	0.017	0.086
	Adiponectin	−0.363	0.158	**0.028**	−0.685	−0.042
**Urinary Cystatin C Interaction with Ox-LDL**						
Model 1 (R^2^ = 0.351) DV: Ox-LDL		Unstandardized Coefficients		P	95% CI for B	
M1	(Constant)	B	SE		Lower Bound	Upper Bound
		179.781	18.072	0.0001	142.969	216.593
	UrCys-C	0.555	0.134	**0.0001**	0.283	0.827
Model 2 (R^2^ = 0.527) DV: Ox-LDL		Unstandardized Coefficients		P	95% CI for B	
M2	(Constant)	B	SE		Lower Bound	Upper Bound
		259.036	28.048	0.0001	201.832	316.239
	UrCys-C	0.459	0.119	**0.0001**	0.216	0.702
	Adiponectin	−0.460	0.135	**0.002**	−0.735	−0.184
Mode 3 (R^2^ = 0.637) DV: Ox-LDL		Unstandardized Coefficients		P	95% CI for B	
M3	(Constant)	B	SE		Lower Bound	Upper Bound
		199.194	31.918	0.0001	134.009	264.379
	UrCys-C	0.382	0.109	**0.001**	0.159	0.605
	Adiponectin	−0.395	0.122	**0.003**	−0.645	−0.146
	TNFα	1.307	0.434	**0.005**	0.421	2.193

A stepwise linear regression analysis was conducted to identify the strongest predictors of association with ox-LDL (dependent variable). Independent predictors (Cys-C, adiponectin, TNFα, NGF, IL8, MCP1, IL1b, IL6, leptin, and insulin) were included in the models (M-3). The stepwise regression criterion for the probability values to enter the models was F ≤ 0.050, and to remove them from models, it was F ≥ 0.100. R^2^ represents the proportion of variance in the dependent variable explained by the independent variable in the linear regression models.

**Table 8 ijms-26-03001-t008:** Mediating factors influencing the relationship between Cys-C and ox-LDL in T2DM patients.

Relationship	Plasma Cystatin C					Urinary Cystatin C				
Mediation Type	Effect (β) (*p*-Value)	SE	t	95% CI (LLCI) (ULCI)	Mediation (%)	Effect (β) (*p*-Value)	SE	t	95% CI (LLCI) (ULCI)	Mediation (%)
Adiponectin										
Total (c)	0.071 (0.0001)	0.012	5.972	(0.047) (0.094)	-	0.443 (0.0004)	0.117	3.771	(0.208) (0.679)	-
Direct (c’)	0.057 (0.0001)	0.012	4.816	(0.033) (0.081)	-	0.347 (0.0022)	0.108	3.213	(0.130) (0.563)	-
Indirect (a*b)	0.013	0.006	-	(**0.003)** **(0.027)**	**18.31** **(PM)**	0.096	0.055	-	(−0.004) (0.212)	21.67 (NS)
Leptin										
Total (c)	0.071 (0.0001)	0.012	5.972	(0.047) (0.094)	-	0.443 (0.0004)	0.117	3.771	(0.208) (0.679)	-
Direct (c’)	0.067 (0.0001)	0.012	5.343	(0.042) (0.092)	-	0.391 (0.023)	0.122	3.200	(0.146) (0.636)	-
Indirect (a*b)	0.004	0.008	-	(−0.003) (0.030)	5.63 (NS)	0.052	0.066	-	(−0.069) (0.194)	11.74 (NS)
TNFα										
Total (c)	0.071 (0.0001)	0.012	5.972	(0.047) (0.094)	-	0.443 (0.0004)	0.117	3.771	(0.208) (0.679)	-
Direct (c’)	0.057 (0.0001)	0.015	3.768	(0.027) (0.087)	-	0.317 (0.065)	0.112	2.830	(0.093) (0.542)	-
Indirect (a*b)	0.014	0.012	-	(−0.010) (0.036)	19.72 (NS)	0.126	0.062	-	**(0.018)** **(0.261)**	**28.44** **(PM)**
Insulin										
Total (c)	0.071 (0.0001)	0.012	5.972	(0.047) (0.094)	-	0.443 (0.0004)	0.117	3.771	(0.208) (0.679)	-
Direct (c’)	0.070 (0.0001)	0.0001	5.931	(0.040) (0.094)	-	0.475 (0.0002)	0.118	4.022	(0.238) (0.712)	-
Indirect (a*b)	0.000	0.002	-	(−0.004) (0.625)	0.00 (NS)	−0.032	0.031	-	(−0.093) (0.032)	0.00 (NS)
IL-6										
Total (c)	0.071 (0.0001)	0.012	5.972	(0.047) (0.094)	-	0.443 (0.0004)	0.117	3.771	(0.208) (0.679)	-
Direct (c’)	0.068 (0.0001)	0.013	5.218	(0.042) (0.094)	-	0.421 (0.0005)	0.114	3.703	(0.193) (0.649)	-
Indirect (a*b)	0.003	0.006	-	(−0.008) (0.016)	4.22 (NS)	0.022	0.033	-	(−0.050) (0.089)	4.99 (NS)
GLP1										
Total (c)	0.071 (0.0001)	0.012	5.972	(0.047) (0.094)	-	0.443 (0.0004)	0.117	3.771	(0.208) (0.679)	-
Direct (c’)	0.067 (0.0001)	0.012	5.366	(0.042) (0.092)	-	0.403 (0.0011)	0.117	3.443	(−0.016) (0.126)	-
Indirect (a*b)	0.004	0.005	-	(−0.003) (0.015)	5.63 (NS)	0.040	0.037	-	(−0.016) (0.013)	9.03 (NS)
IL-1b										
Total (c)	0.071 (0.0001)	0.012	5.972	(0.047) (0.094)	-	0.443 (0.0004)	0.117	3.771	(0.208) (0.679)	-
Direct (c’)	0.070 (0.0001)	0.012	5.843	(0.046) (0.094)	-	0.438 (0.0007)	0.122	3.605	(0.195) (0.682)	-
Indirect (a*b)	0.001	0.001	-	(−0.003) (0.003)	1.41 (NS)	0.005	0.023	-	(−0.057) (0.038)	1.13 (NS)
MCP1										
Total (c)	0.071 (0.0001)	0.012	5.972	(0.047) (0.094)	-	0.443 (0.0004)	0.117	3.771	(0.208) (0.679)	-
Direct (c’)	0.071 (0.0001)	0.012	5.915	(0.047) (0.095)	-	0.443 (0.0005)	0.119	3.734	(0.205) (0.681)	-
Indirect (a*b)	0.0001	0.002	-	(−0.007) (0.003)	1.41 (NS)	0.0003	0.015	-	(−0.036) (0.031)	0.07 (NS)
IL-8										
Total (c)	0.071 (0.0001)	0.012	5.972	(0.047) (0.094)	-	0.443 (0.0004)	0.117	3.771	(0.208) (0.679)	-
Direct (c’)	0.071 (0.0001)	0.012	5.917	(0.095) (0.627)	-	0.447 (0.0004)	0.119	3.745	(0.208) (0.686)	-
Indirect (a*b)	0.0001	0.002	-	(−0.005) (0.002)	1.41 (NS)	-0.004	0.025	-	(−0.079) (0.018)	0.00 (NS)
NGF										
Total (c)	0.071 (0.0001)	0.012	5.972	(0.047) (0.094)	-	0.443 (0.0004)	0.117	3.771	(0.208) (0.679)	-
Direct (c’)	0.074 (0.0001)	0.013	5.825	(0.099) (0.655)	-	0.440 (0.0004)	0.117	3.753	(0.205) (0.675)	-
Indirect (a*b)	−0.003	0.004	-	(−0.013) (0.004)	0.00 (NS)	0.003	0.026	-	(−0.063) (0.049)	0.68 (NS)

The data illustrates the effect of mediating variables on the relationship between Cys-C (independent variable) and ox-LDL (dependent variable). The regression analysis using Hayes PROCESS macro (V4.2) was conducted to generate the results. In a simple mediation (Model 4), the indirect effect (a*b) represents the extent to which the independent variable (Cys-C) influences the dependent variable (ox-LDL) through the individual mediator. The total mediation effect (c) represents the sum of the direct and indirect mediation effects (c’ + a*b). At the same time, LLCI is the lower-level confidence interval, and ULCI is the upper-level confidence interval. A *p* value ≤ 0.05 is considered significant.. Abbreviations: NGF, nerve growth factor; IL-8, interleukin 8; MCP1, monocyte chemoattractant protein-1; IL-1b, interleukin 1b; GLP1, glucagon-like peptide 1; IL-6, interleukin 6; TNFα, tumor necrosis factor-alpha. NS: not significant. PM: partial moderation.

## Data Availability

All data presented are included in the manuscript and Appendix A.

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
