# Peer review of "Adiponectin and TNF-Alpha Differentially Mediate the Association Between Cystatin C and Oxidized LDL in Type 2 Diabetes Mellitus Patients"

_ijms, 2025, doi:10.3390/ijms26073001_

Round 1

Reviewer 1 Report

Comments and Suggestions for Authors

This manuscript presents a correlative analysis of clinical data aimed at elucidating how Adiponectin and TNF-alpha differentially mediate the relationship between Cystatin C and Oxidized LDL in patients with Type-2 Diabetes Mellitus. This perspective is intriguing, as we know that T2D and obesity are multifactorial metabolic disorders. There has been extensive previous research attempting to identify biomarkers for T2D, including adiponectin, TNF-alpha, and some lesser-discussed indicators like Cystatin C. Although previous reports diminish the novelty of this study, I acknowledge that this manuscript is a data-rich clinical correlative analysis based on the data presented.

Oxidized LDL, as an explored marker of "bad" LDL cholesterol's harmful effects, is not widely studied, which adds value to this work. To enhance the presentation and interpretation of data, I recommend the following modifications:

  1. Please display the correlation coefficients (r) and p-values for the scatter plots in Figures 1D-F.
  2. Convert Table 2 into a heatmap for a more intuitive understanding and consider relocating this table to the supplementary materials.
  3. Convert Table 3 into a heatmap for clearer visualization, and similarly, relocate this table to the supplementary materials.
  4. The tables throughout the manuscript somewhat reduce its readability. I suggest transforming the content of these tables into figures where possible.
  5. In the introduction, please include information on the proportion of Oxidized LDL relative to total LDL.

These suggestions are intended to enhance the clarity and impact of the findings, facilitating a better understanding of the complex interactions in T2D patients.

Comments on the Quality of English Language

The quality of English expression is acceptable.

Reviewer 2 Report

Comments and Suggestions for Authors

The authors have created a very important summary of the mechanism between Cys-C level and ox-LDL, taking into account numerous parameters obtained from the blood of patients with type 2 diabetes. Below are some comments worth considering.

1. In line 434, the author writes about 57 patients with diabetes, while in line 450 there is information about freezing samples from patients with type 2 diabetes and from healthy subjects.

2. It is worth including a Limitation section in which the limitations of this study are described, e.g. small study group, no control group, no information about medications taken, testing only a part of the entire network of adipokines and inflammatory cytokines.

3. I suggest adding "preliminary results" to the title.

4. The study methods are missing a sentence or two about collecting urine samples from subjects and a more detailed description of BMI in the study group with a description of the number in each class according to WHO standards.

Round 2

Reviewer 1 Report

Comments and Suggestions for Authors

The author has solved the problems I raised well, and I suggest acceptance.

Comments on the Quality of English Language

The author's manuscript is generally understandable in English, but it could be improved.